# Language Guidance for Supervised Vision Training: An Empirical Study of Generalization

## Abstract

Deep neural networks have achieved remarkable success on vision benchmarks, yet they continue to struggle with many generalization challenges. Supervised vision training relies on one-hot labels, which provide limited information about semantic structure and shared attributes between classes. This limited supervision can leave visual representations vulnerable to distribution shifts, spurious correlations, texture bias, adversarial perturbations, and forgetting in sequential learning settings. We study whether pretrained language models can serve as lightweight auxiliary supervision for vision training without requiring paired image-text data, prompt engineering, or contrastive objectives. Specifically, we evaluate two forms of language guidance, Explicit Language Guidance (ExLG) and Implicit Language Guidance (ImLG). We conduct a comprehensive evaluation across six generalization regimes, including in-distribution, out-of-distribution generalization, shortcut and spurious correlation resistance, texture and shape bias, adversarial robustness, and continual learning. Our analyses show that the two mechanisms have complementary strengths, with explicit guidance consistently benefiting in-distribution, low-data performance, and continual learning retention, while implicit guidance is often more useful in shortcut-sensitive settings and under stronger adversarial perturbations. Importantly, both are lightweight and add minimal parameters and training overhead. The analyses characterize when language-derived structure helps supervised vision training and provides a practical roadmap for using off-the-shelf pretrained models from another modality as auxiliary supervision.

## 1 Introduction

Despite remarkable progress on standard vision benchmarks, deep neural networks (DNNs) often exhibit limited generalization beyond the training distribution. These failures appear in several forms, including brittleness under distribution shift, susceptibility to shortcut learning (Geirhos et al., 2020; Jo & Bengio, 2017), texture bias (Geirhos et al., 2018) and adversarial perturbations, where models rely on spurious correlations, local textures or noise in the data rather than learning the true underlying causal patterns. These challenges are further amplified in continuous learning (Parisi et al., 2019), where the data distribution changes over time and the models must acquire new knowledge without forgetting previously learned tasks. Although these settings are usually studied separately, they expose a shared limitation. Visual representations learned from conventional supervision can fit the training distribution while remaining weakly constrained for broader generalization.

A central factor underlying these limitations is the nature of the supervisory signal. A one-hot label identifies the target class, but it does not describe semantic attributes, relations between classes, or higher-level conceptual structure. As a result, the vision encoder must infer useful abstractions from pixel statistics alone, which can encourage reliance on superficial cues rather than meaningful abstractions. Language offers a lightweight source of additional structure because class names and descriptions encode attributes, conceptual relations, and shared structure that are absent from one-hot labels.

Pretrained language models provide a practical way to use this structure without collecting paired image-text data or changing the task into a multi-modal retrieval problem. Figure 1 illustrates motivation in a

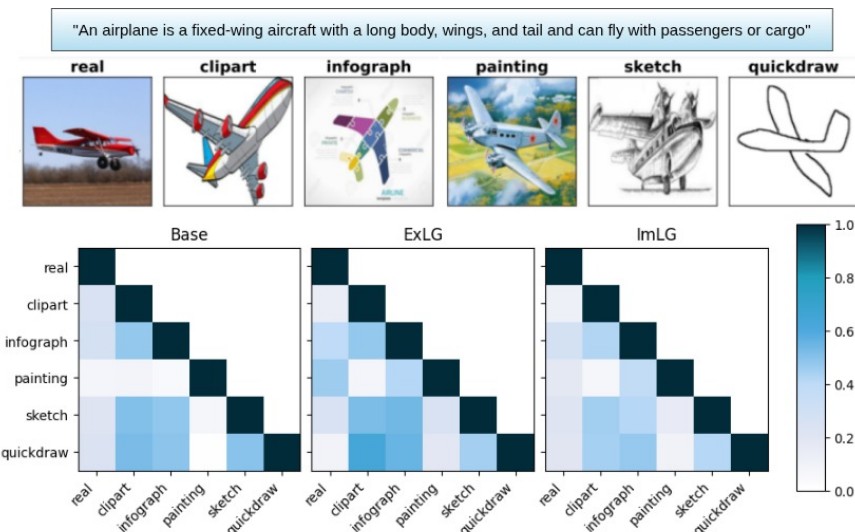

Figure 1: Similarity analysis to show if language context introduced by Explicit or Implicit Language Guidance enhances semantic alignment across domains in DN4IL dataset.

domain generalization setting. On DN4IL dataset with six challenging domains, a vision-only model shows low cross-domain feature similarity, especially for visually challenging domains such as paintings, sketches, and quickdraw. With language guidance, cross-domain similarity increases, suggesting that language-derived supervision can alter the geometry of visual representations. This motivates our central question. Can off-the-shelf frozen language models provide useful auxiliary supervision for supervised vision training when learning is restricted to image-label pairs.

We study this question through two lightweight mechanisms. Explicit Language Guidance (ExLG) directly aligns visual representations with class-level language descriptions using a similarity-preserving objective. Implicit Language Guidance (ImLG) inserts a frozen language block into the vision pipeline, allowing language-derived structure to influence the features passed to the classifier in a more implicit manner. Both mechanisms are modular and avoid paired image-text data, contrastive multi-modal pretraining, and prompt-based inference. Our goal is not to introduce a new learning paradigm, but to provide an empirical characterization of when language guidance helps, when it has limited effect, and what trade-offs arise from explicit versus implicit integration. We evaluate ExLG and ImLG across in-distribution, out-of-distribution transfer, shortcut-sensitive datasets, texture/shape bias, adversarial robustness, and continual learning. Across these settings, explicit guidance is more consistently useful for clean accuracy, low-data performance, and retention in continual learning, while implicit guidance is often more competitive in settings dominated by shortcut cues or stronger adversarial perturbations.

- We study two lightweight and modular forms of language guidance for supervised vision training that do not rely on paired image-text data or multi-modal pretraining.

- We evaluate these mechanisms across multiple generalization regimes, including out-of-distribution transfer, shortcut reliance, texture bias, adversarial robustness, and continual learning.

- We show that explicit and implicit language guidance induce different trade-offs. ExLG is more consistently beneficial for clean generalization and retention, while ImLG is often stronger in some robustness-oriented settings.

- We provide practical guidance for using pretrained off-the-shelf models from another modality as auxiliary sources of structure in supervised vision pipelines.

## 2 Related Works

Vision-Language Models (VLMs) integrate vision and language representations for tasks such as visual question answering, captioning, and retrieval. They typically employ methods that align image and text embeddings via contrastive learning on large-scale multi-modal datasets (Radford et al., 2021), or fuse modalities to enable complex reasoning and captioning (Li et al., 2022). Some models enhance multi-modal understanding through instruction tuning (Liu et al., 2024). These models have shown strong transfer capabilities, but their standard use of language differs from the setting studied here. They typically require paired image-text data, formulate classification as text-image retrieval, or rely on prompt-based inference and fine-tuning when adapting to new visual domains.

Beyond complete pretraining in VLMs, several studies have begun to leverage language to refine vision models. Caption-based contrastive methods use language to select positive and negative pairs more precisely than image augmentations alone (El Banani et al., 2023). Other approaches introduce proxy objectives that combine visual inputs with text-derived targets, such as predicting image tags from captions or using image-conditioned masked language modeling (Sariyildiz et al., 2020). Similarly, (Merullo et al., 2022) explore simple linear mappings from vision features to text embedding space for cross-domain retrieval tasks. These methods generally depend on metadata, proxy objectives, or paired text supervision, and are often designed for retrieval, captioning, or multi-modal generation rather than classic supervised vision training with image-label pairs.

Recent analyses provide a broader motivation for cross-modal guidance. The Platonic Representation Hypothesis argues that representations learned across different modalities may converge toward shared high-level abstractions (Huh et al., 2024), and empirical work has studied conceptual alignment between vision and language representations (Maniparambil et al., 2024). Text transformers can also exhibit neurons that respond to semantically related visual and textual inputs (Schwettmann et al., 2023). However, these studies do not specify how, when and in what setting such a structure should be incorporated into a supervised vision pipeline. What remains underexplored is whether a single lightweight intervention using language guidance via an off-the-shelf PLM can address multiple vision generalization failures within a standard supervised pipeline, and under what conditions each form of guidance is most effective.

## 3 Language Guidance for Supervised Vision Training

### 3.1 Motivation

We revisit the role of **Inductive Biases** in neural networks. Humans learn high-level abstractions facilitated by language and can generalize robustly across diverse contexts and related arguments in representation learning suggest that language can make high-level concepts available as reusable structure (Goyal & Bengio, 2022). Our explicit and implicit mechanisms are inspired by cognitive theories that distinguish between System 2, which is associated with explicit and verbalizable processing, and System 1, which is associated with implicit and intuitive processing (Kahneman, 2011). Explicit knowledge can be stated, communicated and used as a direct source of guidance. Instead, implicit processing shapes behavior through internal transformations that are not expressed as explicit rules. These perspectives motivate two questions. Can structured knowledge from pretrained language models serve as a lightweight inductive bias for supervised vision training? If so, how should this knowledge be used to guide visual representations?

### 3.2 Setup and Design Principles

We consider a standard supervised vision setting with training samples $\{(x_i, y_i)\}_{i=1}^{N}$, where $x_i$ is an image and $y_i \in \{1, \ldots, C\}$ is a class label. A vision encoder $g_\theta$ maps each image to a feature vector $z_i = g_\theta(x_i)$ and a classifier $h_\phi$ maps the feature vector to class logits $\hat{y}_i = h_\phi(z_i)$. The baseline model is trained with the standard cross-entropy objective

$$\mathcal{L}_{\text{cls}} = -\frac{1}{N} \sum_{i=1}^{N} \log \frac{\exp(\hat{y}_{i,y_i})}{\sum_{c=1}^{C} \exp(\hat{y}_{i,c})}. \tag{1}$$

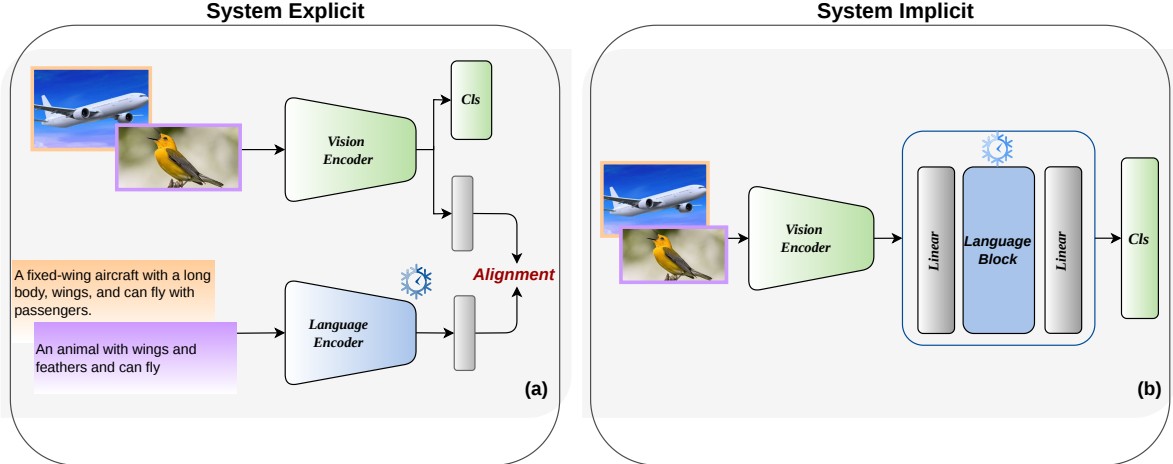

Figure 2: (a) Explicit Language Guidance: Learning visual representations with explicit supervision from language descriptions. (b) Implicit Language Guidance: Learning visual representations via an embedded frozen language block for implicit supervision.

The language model is frozen and is used as an auxiliary source of structure, and the two mechanisms differ in the way the language enters the pipeline (Figure 2). Explicit Language Guidance (**ExLG**) adds a direct alignment objective by aligning visual representations with the semantic structure induced by class-level language descriptions. Implicit Language Guidance (**ImLG**) introduces language-derived structure more indirectly by inserting a frozen language block into the vision pipeline.

### 3.3 Explicit Guidance

Explicit Language Guidance utilizes explicit information such as language descriptions of the classes to guide the training process, as one-hot labels specify class identity but provide limited relational or semantic structure during training. Let $l_c$ denote the language description for class $c$. A frozen pretrained language model $plm$ maps this text to a language embedding

$$e_c = plm(l_c). \tag{2}$$

The image encoder is optimized with the usual classification loss and an auxiliary alignment loss that encourages the visual feature space to preserve the relational structure of the language feature space. The language model is used only during training, so ExLG does not add inference-time language computation.

Rather than directly matching individual embeddings, we use a similarity-preserving objective that encourages the relational structure of the visual feature space to resemble that induced by the language embeddings. **Similarity Preserving Loss** works by computing pairwise similarity matrices from the activation maps of both the vision and language models. The loss function penalizes differences between these similarity matrices, encouraging the vision model to learn representations that are aligned with the semantic knowledge embedded in the language descriptions (more details in the Appendix).

The overall loss function is defined as:

$$\mathcal{L} = \mathcal{L}_{\text{cls}} + \lambda \mathcal{L}_{\text{align}} \tag{3}$$

where $\mathcal{L}_{\text{cls}}$ is the classification loss, $\mathcal{L}_{\text{align}}$ is the alignment loss, and $\lambda$ controls the influence of the alignment term.

Let $F_v = [z_1, \ldots, z_N]^\top$ denote the visual feature matrix for a batch and $F_l = [e_{y_1}, \ldots, e_{y_N}]^\top$ denote the corresponding language feature matrix. Let $\bar{F}_v$ and $\bar{F}_l$ denote row-normalized versions of these matrices. To

Table 1: Summary of all analyses and datasets in this study.

| Vision Encoder | ResNet18 | ResNet50 | ViT | |
|---|---|---|---|---|
| PLM | Sentence Transformer variants | CLIP | | |
| **Analysis** | **Datasets** | | | |
| IID | CIFAR10 | CIFAR100 | TinyImageNet | ImageNet-100 |
| OOD | ImageNet-O | ImageNet-R | ImageNet-A | |
| Shortcut Learning | Tinted-CIFAR10 | Skewed-CelebA | WaterBirds | |
| Texture Bias | Stylized TinyImageNet | Texture-Shape Cue-Conflict | | |
| Adversarial Robustness | CIFAR10 | | | |
| Continual Learning | Seq-CIFAR10 | Seq-TinyImageNet | DN4IL | |

guide the vision encoder toward the pairwise structure induced by language, we define

$$\mathcal{L}_{\mathrm{align}} = \frac{1}{N^2}\|\mathcal{S}_v - \mathcal{S}_l\|^2 \tag{4}$$

$$\mathcal{S}_v = \bar{F}_v\bar{F}_v^\top, \quad \mathcal{S}_l = \bar{F}_l\bar{F}_l^\top \tag{5}$$

where $\mathcal{S}_v$ and $\mathcal{S}_l$ are the pairwise similarity matrices for the vision and language features. The goal is to align the similarity structure in the visual embedding space with the similarity structure in the language embedding space. This formulation requires only class-level descriptions rather than instance-level text, making ExLG lightweight in terms of additional data requirements.

### 3.4 Implicit Guidance

Implicit language guidance architecturally introduces the language-derived structure, inserting a frozen language encoder block between the vision encoder and the classifier (Figure 2). The motivation is that a pretrained language block may act as a fixed transformation that reshapes visual features according to the structure learned from the language. This differs from ExLG in two ways. First, the language model influences the feature pathway directly rather than providing a separate target. Second, the supervision remains entirely classification-based, since no language alignment loss is added. In this sense, ImLG provides an implicit form of guidance and changes the processing route available to the classifier while keeping the language component frozen.

Let *lb* denote a frozen block extracted from a pretrained language model. Since the dimension of visual features may differ from the dimension of the language block, we use trainable projection layers $p_1$ and $p_2$. The ImLG feature pathway is

$$u_i = p_1(g_\theta(x_i)), \tag{6}$$

$$\tilde{z}_i = p_2(lb(u_i)), \tag{7}$$

and the classifier predicts

$$\hat{y}_i = h_\phi(\tilde{z}_i). \tag{8}$$

The model is trained with the same cross-entropy loss $\mathcal{L}_{\mathrm{cls}}$, while the language block $b$ remains frozen. Only the vision encoder, projection layers, and classifier are updated. This alignment allows the vision encoder to utilize the rich semantic embeddings generated by the PLM without the need for regularization or additional complex loss functions. The classifier then operates on these filtered or semantically enriched features to perform visual recognition tasks.

## 4 Empirical Study

We systematically evaluate both language guidance mechanisms across six generalization regimes: in-distribution accuracy and sample efficiency, out-of-distribution transfer, shortcut and spurious correlation

Table 2: In-distribution accuracy and CIFAR-10 sample efficiency. ExLG is most consistent in clean and low-data settings.

| METHOD | IID | | | CIFAR-10 DATA EFFICIENCY (%) | | | |
|---|---|---|---|---|---|---|---|
| | CIFAR-10 | CIFAR-100 | TINYIMGNET | 2 | 5 | 10 | 20 |
| BASELINE | $94.84_{\pm0.14}$ | $76.98_{\pm0.39}$ | $58.73_{\pm0.35}$ | $45.71_{\pm1.52}$ | $55.42_{\pm1.08}$ | $67.04_{\pm2.19}$ | $79.62_{\pm2.60}$ |
| ExLG | $\mathbf{95.12_{\pm0.05}}$ | $\mathbf{77.59_{\pm0.08}}$ | $\mathbf{65.63_{\pm0.26}}$ | $\mathbf{47.88_{\pm0.53}}$ | $\mathbf{57.24_{\pm1.95}}$ | $\mathbf{69.97_{\pm1.87}}$ | $\mathbf{84.75_{\pm0.61}}$ |
| IMLG | $93.41_{\pm0.46}$ | $74.10_{\pm0.91}$ | $60.02_{\pm0.16}$ | $45.03_{\pm2.06}$ | $55.53_{\pm2.06}$ | $67.82_{\pm1.42}$ | $79.03_{\pm0.57}$ |

Table 3: Out-of-distribution transfer to ImageNet variants.

| METHOD | IMAGENET-O | IMAGENET-R | IMAGENET-A |
|---|---|---|---|
| BASELINE | $41.73_{\pm1.45}$ | $10.59_{\pm0.41}$ | $1.92_{\pm0.53}$ |
| ExLG | $\mathbf{46.70_{\pm1.02}}$ | $\mathbf{14.95_{\pm0.07}}$ | $\mathbf{2.94_{\pm0.33}}$ |
| IMLG | $42.20_{\pm0.81}$ | $12.10_{\pm0.17}$ | $2.37_{\pm0.27}$ |

Table 4: ImageNet-100 results with larger CNN and ViT backbones.

| | Baseline | ExLG | ImLG |
|---|---|---|---|
| ResNet50 | 71.46 | **80.56** | 72.37 |
| ViT | 54.77 | **56.16** | 55.06 |

resistance, texture and shape bias, adversarial robustness, and continual learning. We experiment on both CNNs and transformers, and the different architectures and datasets used are tabulated in Table 1. The appendix includes more results and ablations with different sizes of PLMs, shown in Tables 10 and 11. Detailed experimental setups and hyper-parameters are provided in the appendix.

## 4.1 IID, Sample Efficiency, and OOD Transfer

We first evaluate whether language guidance improves in-distribution performance and transfer to shifted test distributions. For clean accuracy, we use CIFAR-10, CIFAR-100, TinyImageNet, and ImageNet-100. For sample efficiency, we train on progressively smaller fractions of CIFAR-10. For the OOD evaluation, we assess the models' robustness on challenging benchmarks derived from the ImageNet dataset, namely ImageNet-O (which contains outlier data points), ImageNet-R (comprising artistic renditions of objects), and ImageNet-A (which contains adversarially filtered images) (Hendrycks et al., 2021a;b).

Table 2 shows that ExLG is the most reliable mechanism for clean and low-data settings. It improves CIFAR-10, CIFAR-100, and TinyImageNet accuracy, and the gains become more pronounced when fewer CIFAR-10 training examples are available. Table 3 shows that language guidance also improves OOD transfer from TinyImageNet to ImageNet variants. ExLG gives the strongest gains in these OOD benchmarks, while ImLG also improves over the baseline. The same pattern appears when scaling to ImageNet-100 with larger CNN and transformer backbones (Table 4). Notably, ExLG outperforms ImLG in these settings, indicating that direct language supervision provides more utility in in-distribution testing.

## 4.2 Shortcut Learning

Shortcut learning is a common problem in neural networks, where models rely on superficial patterns or spurious correlations present in the training data to make predictions (Geirhos et al., 2020). To test the extent of shortcut learning, we employ three specially curated datasets: **(1) Tinted-CIFAR10** is a variant of CIFAR10, where a unique color tint is added to each class. **(2) Skewed-CelebA** is a skewed version of the CelebA (Liu et al., 2015) dataset with a spurious association between gender and hair color. The training data are heavily biased towards samples of blonde women and non-blonde men, while test data have non-blonde women and blonde men samples. **(3) Waterbirds** (Sagawa et al.) creates a spurious association between bird type and background. Landbirds are set against land backgrounds, and vice versa. At test time, the models are evaluated on birds placed in conflicting environments.

Table 5 shows that the baseline model performs poorly across all three datasets. Language guidance improves performance over the baseline, particularly on Skewed-CelebA and Waterbirds, where shortcut cues strongly influence the baseline. With explicit language guidance, the model significantly improves its performance.

Table 5: Shortcut learning on Tinted-CIFAR10, Skewed-CelebA, and Waterbirds. Language-guided models improve robustness to spurious features and conflict groups.

| Method | Tint-CIF10 | Skewed-CelebA | | | | |
|---|---|---|---|---|---|---|
| | | Overall | NonBlonde-M | Blonde-F | Blonde-M | NonBlonde-F |
| Baseline | 16.45±1.81 | 61.28±1.21 | 94.71±0.08 | 92.21±1.02 | 56.38±0.39 | 27.74±2.29 |
| +ExLG | **18.24±0.60** | **72.11±1.28** | **96.29±0.30** | **95.18±0.31** | **68.33±0.98** | **47.67±2.31** |
| +ImLG | **18.51±1.04** | **75.90±1.79** | **97.85±0.65** | **96.81±0.11** | **69.77±1.03** | **53.84±3.38** |
| | | WaterBirds | | | | |
| Method | | Overall | Landbird/land | Waterbird/water | Landbird/water | Waterbird/land |
| Baseline | | 62.60±0.13 | 95.29±0.45 | 76.95±2.15 | 40.77±0.60 | 12.53±2.59 |
| +ExLG | | **64.34±0.32** | **96.89±0.45** | **83.45±2.18** | **47.64±0.75** | **18.16±2.46** |
| +ImLG | | **65.22±0.29** | **96.68±0.14** | **86.84±2.62** | **46.13±0.91** | **20.16±3.79** |

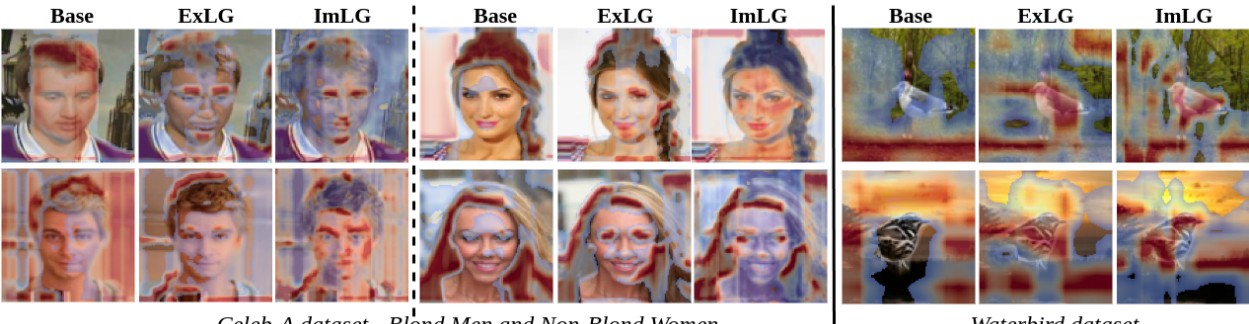

Figure 3: Activation maps of the models on the Skewed-CelebA and Waterbird dataset. Language-guided models focus on the salient features, while conventional methods focus on spurious cues (hair color).

In particular, the ExLG model improves classification by 22% for blonde males and 70% for non-blonde females—categories unseen during training. ImLG achieves even greater gains, with an increase of 94% for non-blonde females and of 66% in the category of waterbird-land of the Waterbirds dataset.

To further gain insight into this behavior, we use Grad-CAM (Selvaraju et al., 2017) to generate visual maps. Figure 3 shows that the baseline often attends to shortcut cues such as hair color or background. Language-guided models prioritize more essential attributes, such as facial features in Skewed-CelebA and toward the bird in Waterbirds. These maps suggest that the language block can act as a conceptual filter, encouraging the vision model to spread its attention towards relevant information and ignoring misleading elements such as background or textures.

### 4.3 Texture vs Shape Bias

Deep neural networks often exhibit a strong reliance on texture information to make predictions (Geirhos et al., 2018). To systematically analyze this phenomenon, we conducted two distinct experiments. **(1) Stylized-TinyImageNet** - We applied style transfer (Huang & Belongie, 2017) on the TinyImageNet dataset, with various texture patterns, while keeping the underlying object shapes intact. The stylization alpha ($\alpha$) determines the extent to which the original image's texture is replaced with the style features from a reference image. Few examples are shown in Figure 8 **(2) Texture-Shape Cue Conflict** - We designed a modified TinyImageNet dataset featuring a texture-shape cue conflict, where the object shape belongs to one category while the texture is borrowed from another. (e.g., an image shaped like a car but textured like a keyboard).

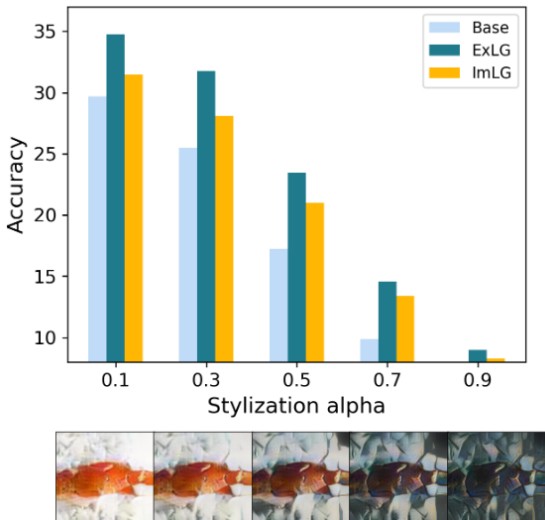
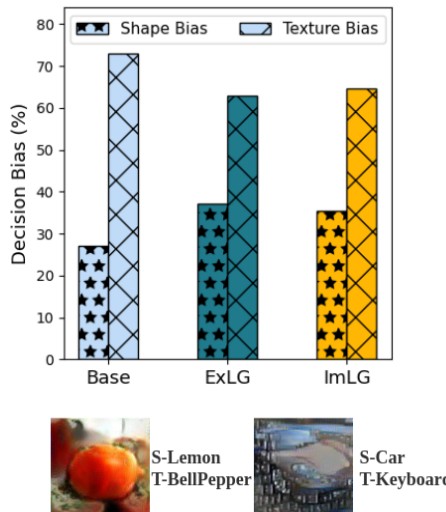

Figure 4: (1) Texture Bias - Analysis on Stylized TinyImageNet across three levels of stylization. (2) Shape vs Texture bias on a conflict cue dataset that has different textures applied on objects

Figure 4 (left) shows that both ExLG and ImLG are more robust than the baseline with increasing stylization. Figure 4 (right) shows that standard vision encoders predominantly prioritize texture cues while language-guided models exhibit a marked shift towards shape-based classification, significantly reducing texture reliance. Language guidance appears to facilitate the learning of more abstract and global representations, potentially improving model generalization when faced with texture variations or misleading cues. Research suggests that unlike neural networks, which often prioritize texture, humans generally exhibit a stronger bias towards shape (Geirhos et al., 2018; Gowda et al., 2022) and the introduction of language guidance nudges models towards this human-like perception.

### 4.4 Robustness

DNNs, despite their proficiency in pattern recognition, remain highly susceptible to adversarial attacks, which involve subtle, often imperceptible alterations to input data that drastically skew model outputs (Szegedy et al., 2013). These adversarial perturbations, which humans typically resist, expose a critical vulnerability in DNNs, underscoring their potential risk in practical applications. We evaluate robustness using projected gradient descent attacks (PGD) on CIFAR-10 in varying attack strengths $\epsilon$ (Madry et al., 2018).

As shown in Figure 5, the ExLG model consistently exceeds the baseline model (Base-Cls) at all levels of attack strength, demonstrating stronger adversarial robustness. However, the interesting observation comes from the behavior of the ImLG model. Although its performance is lower than both ExLG and Baseline at lower attack strengths, it becomes significantly more robust as the attack strength increases. For higher attack magnitudes, ImLG outperforms both ExLG and Baseline, showcasing superior resilience to stronger attacks.

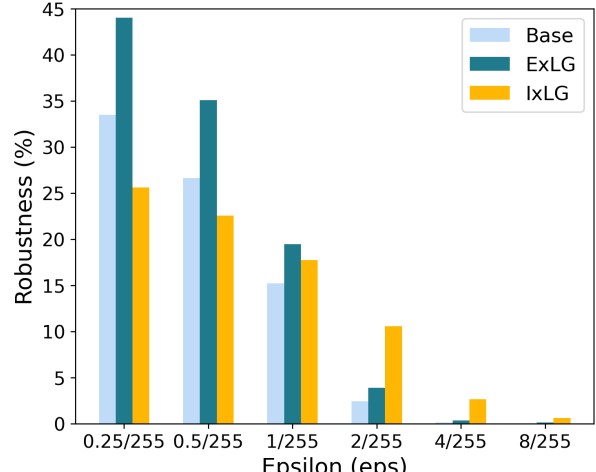

Figure 5: Robustness against PGD-10 adversarial attack on varying strengths ($\epsilon$) on the CIFAR-10

This behavior aligns with the hypothesis Jo & Bengio (2017), that models capable of learning high-level

| | Class-IL | | Task-IL | Domain-IL |
|---|---|---|---|---|
| Method | Seq-CIF10 | Seq-Tiny | Seq-CIF10 | DN4IL |
| SGD | $19.62_{\pm0.05}$ | $7.92_{\pm0.26}$ | $61.02_{\pm3.33}$ | $20.83_{\pm0.24}$ |
| Joint | $92.20_{\pm0.15}$ | $59.99_{\pm0.19}$ | $98.31_{\pm0.12}$ | $59.93_{\pm1.07}$ |
| ER | $44.79_{\pm1.86}$ | $18.38_{\pm0.16}$ | $91.19_{\pm0.94}$ | $24.15_{\pm0.34}$ |
| +ExLG | $\mathbf{54.84_{\pm0.97}}$ | $\mathbf{20.39_{\pm0.15}}$ | $\mathbf{93.84_{\pm0.85}}$ | $\mathbf{27.71_{\pm0.64}}$ |
| +ImLG | $\mathbf{47.57_{\pm0.20}}$ | $\mathbf{19.86_{\pm0.24}}$ | $91.89_{\pm0.74}$ | $\mathbf{24.22_{\pm0.12}}$ |
| ER-ACE | $65.21_{\pm0.69}$ | $21.60_{\pm1.58}$ | $93.24_{\pm0.12}$ | $26.76_{\pm0.72}$ |
| +ExLG | $\mathbf{66.42_{\pm0.18}}$ | $\mathbf{23.38_{\pm0.21}}$ | $\mathbf{93.54_{\pm0.42}}$ | $\mathbf{27.77_{\pm0.12}}$ |
| +ImLG | $\mathbf{65.94_{\pm0.09}}$ | $\mathbf{22.06_{\pm0.57}}$ | $91.46_{\pm0.68}$ | $25.45_{\pm0.46}$ |
| o-EWC | - | - | $66.98_{\pm3.20}$ | $19.94_{\pm0.24}$ |
| +ExLG | - | - | $\mathbf{72.54_{\pm2.95}}$ | $\mathbf{22.70_{\pm0.25}}$ |
| +ImLG | - | - | $\mathbf{69.70_{\pm1.95}}$ | $17.64_{\pm0.29}$ |
| SI | - | - | $69.69_{\pm2.86}$ | $19.40_{\pm1.71}$ |
| +ExLG | - | - | $\mathbf{72.22_{\pm1.99}}$ | $\mathbf{23.35_{\pm2.16}}$ |
| +ImLG | - | - | $\mathbf{71.10_{\pm1.79}}$ | $17.06_{\pm0.14}$ |

Table 6: Class-, Task- and Domain-Incremental learning in CL

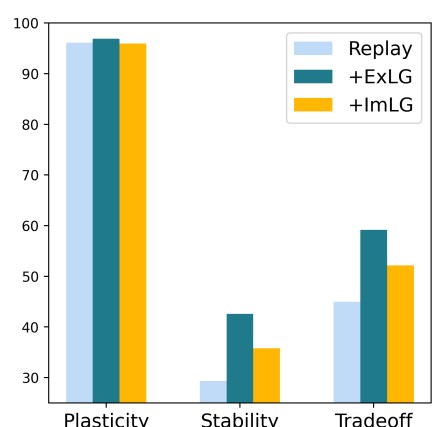

Figure 6: Plasticity-Stability

abstractions should exhibit increased robustness to data perturbations. Therefore, the integration of language guidance not only enhances test performance, but also facilitates the development of more robust representations.

## 4.5 Continual Learning

Continual Learning (CL) (Parisi et al., 2019) focuses on the challenge of learning new tasks sequentially without forgetting previously learned tasks, a phenomenon known as catastrophic forgetting. Class-incremental learning (Class-IL) introduces new classes with each task, while Task-IL, similar to Class-IL, has access to task identifiers during inference. Domain-incremental learning (Domain-IL) presents the same classes across various domains, addressing domain shifts. All three settings test the model's ability to generalize and prevent forgetting (Van de Ven & Tolias, 2019). Our experiments utilize Experience Replay (ER) (Buzzega et al., 2020) and ER-ACE (Caccia et al., 2022) as replay-based methods to combat forgetting by replaying samples from a fixed buffer. We also use two regularization methods, Online Elastic Weight Consolidation (oEWC) (Schwarz et al., 2018) and Synaptic Intelligence (SI) (Zenke et al., 2017), which ensure critical weights remain unchanged without need for memory samples.

Table 6 shows that the ExLG models consistently outperform across all tasks, with significant gains in challenging Class-IL, achieving a 22% improvement with experience replay. ExLG also improves Domain-IL performance on DN4IL dataset, which has six challenging domains, across all methods. ImLG provides some gains with replay methods, but is less consistent with regularization methods. This pattern suggests that direct semantic supervision is more reliable in preserving discriminative structure in sequential tasks than indirect feature-pathway guidance. Overall, these results highlight that explicit language supervision through textual information enhances long-term retention and stability in continual learning. A detailed breakdown of task-wise performance on Seq-CIFAR10 is depicted in Figure 9.

**Plasticity-Stability Trade-off** is crucial in continual learning, reflecting the balance between acquiring new knowledge (plasticity) and retaining old information (stability). As shown in Figure 6, while the Base model demonstrates high plasticity, it suffers from low stability. In contrast, the language-guided models (ExLG and ImLG) exhibit higher stability, suggesting their effectiveness in maintaining balance, enabling them to adapt to new tasks without losing significant information from previous ones.

## 5   Discussion

We investigate how language-derived supervision can be incorporated into supervised vision training, which traditionally relies only on one-hot labels. The empirical results show that explicit and implicit language guidance leads to different performance profiles across evaluation settings. Explicit Language Guidance (ExLG) is more consistently beneficial for clean accuracy, performance in limited-data settings, and retention in continual learning. In contrast, Implicit Language Guidance (ImLG) provides greater gains in settings where performance is strongly affected by shortcut features, spurious correlations, or stronger adversarial perturbations. Both methods reduce texture bias and encourage shape-based reasoning—a more human-aligned approach to visual perception.

These findings suggest that the effect of language guidance depends not only on whether language is introduced but also on how it is incorporated into training. Direct alignment to class-level language descriptions provides a stable auxiliary objective that is especially useful with preserving discriminative structure. Indirect integration through a frozen language block appears to alter feature use in ways that are more helpful when robustness to superficial cues matters. The results on transformer backbones further suggest that the benefit of language guidance may depend on the representational capacity of the underlying architecture, with smaller gains observed when self-attention already captures part of the relevant structure.

**Guidelines:** This paper is intended as a controlled empirical study and the central contribution is a technically grounded account of how different forms of language guidance affect different aspects of vision generalization and robustness. The findings provide practical guidance for practitioners to apply language supervision in standard vision pipelines. When the goal is strong in-distribution accuracy, sample efficiency, or stable retention across sequential tasks, explicit alignment to class-level language descriptions is the more reliable option. When the target setting is especially sensitive to superficial correlations or requires robustness under more severe perturbations, implicit language-conditioned structure may offer greater benefit. In both cases, the methods remain lightweight relative to multi-modal pretraining. For example, Table 12 in the appendix shows that ResNet18 with language guidance achieves higher accuracy than the ResNet50 baseline while using fewer trainable parameters and substantially less training time. Our findings serve as a roadmap for leveraging foundation models to address vision-centric generalization challenges.

**Limitations:** This study focuses on a controlled evaluation of lightweight language guidance in a standard vision supervised training setup across multiple generalization regimes. The experiments are therefore intended to characterize the behavior and trade-offs of explicit and implicit guidance of one modality on other, rather than a novel method. The evaluation is limited to classification-oriented settings, and future work could study whether vision encoders trained with language guidance also improve downstream tasks such as detection and segmentation.

## 6   Broader Impact

Beyond the specific methods studied here, this work has a broader relevance for research on cross-modal supervision. In particular, it supports the view that pretrained models from one modality can provide useful inductive structure for learning in another modality. Instead of using a pretrained language model only for language tasks or as part of a fully multi-modal system, we study whether it can guide a vision model trained from ordinary image-label data. This suggests a broader research direction in which pretrained models are used not only for direct transfer within their native modality, but also as auxiliary sources of structure for representation learning across modalities. The study provides evidence that standard pretrained models can be used as controlled tools to investigate what types of information transfer between modalities and how different forms of cross-modal integration affect learned representations. This may help motivate future work on using pretrained models from language, audio, vision, or other domains to shape supervision in settings where direct annotations are limited or where standard label-based supervision is structurally weak. A useful broader impact of studies like this one is therefore methodological, and they can help establish more principled ways of evaluating when cross-modal supervision is beneficial, what trade-offs it introduces, and how it should be incorporated into learning systems.

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

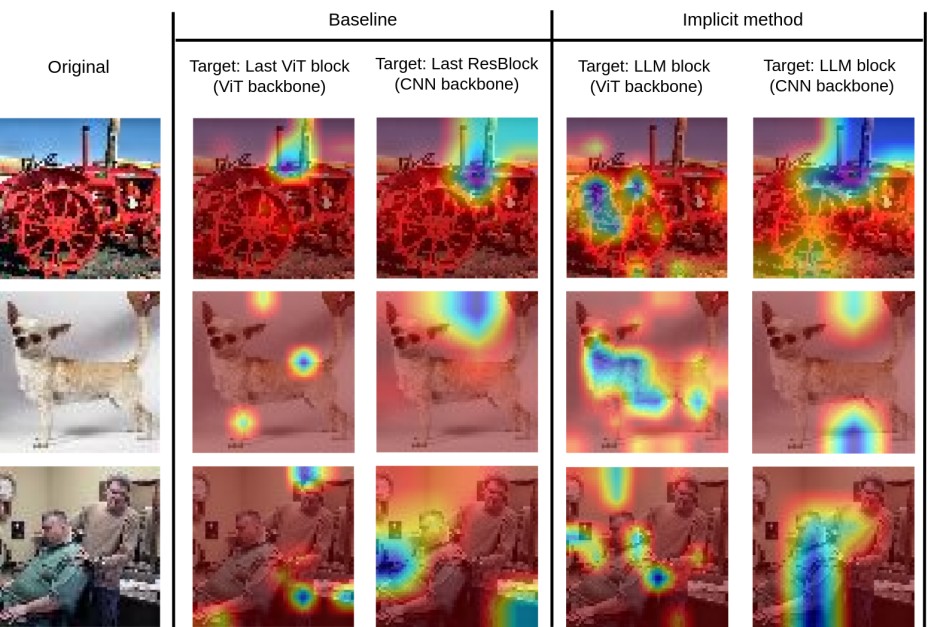

Figure 7: Comparison of activation maps after each block without (**left**) and with (**right**) implicit language-guided training.

# A   Appendix

# B   Explicit-Implicit Analysis

## B.1   CKA Analysis

### B.1.1   CKA

Centered Kernel Alignment (CKA) is a widely used method for measuring the similarity between two representations in neural networks. It quantifies how well these representations align, allowing us to compare features learned by different layers or models. CKA is computed using dot products of representations in the form of Gram matrices, which capture pairwise similarities between examples in each representation. By comparing these Gram matrices, CKA evaluates the structural alignment between two sets of representations, making it particularly useful for understanding the alignment between the activations of a vision encoder and a language model.

The CKA similarity between two feature matrices, $X \in \mathbb{R}^{n \times d_1}$ and $Y \in \mathbb{R}^{n \times d_2}$, where $n$ is the number of samples and $d_1$ and $d_2$ are the dimensions of the features, is calculated as follows. First, we compute the centered Gram matrices $K$ and $L$ for $X$ and $Y$, respectively:

$$\mathrm{CKA}(X, Y) = \frac{\mathrm{HSIC}(K, L)}{\sqrt{\mathrm{HSIC}(K, K) \cdot \mathrm{HSIC}(L, L)}} \tag{9}$$

Here, HSIC (Hilbert-Schmidt Independence Criterion) measures the similarity between the two Gram matrices:

CKA is invariant to orthogonal transformations and isotropic scaling of the representations, making it a robust tool for comparing representations between models. By using CKA, we can effectively evaluate how well representations learned by a vision encoder align with those of a language model, providing deeper insights into cross-modal learning and feature alignment.

For this analysis, we utilize DN4IL dataset, specifically designed for domain adaptation. The dataset has six domains - Real, Clipart, Infograph, Painting, Sketch and Quickdraw. Base utilizes a standard ResNet18

Table 7: Hyperparameters: All models are trained for 100 epochs using the SGD optimizer, except for implicit methods, which employ the AdamW optimizer due to the inclusion of a transformer block.

| Method | CIFAR10, CIFAR100, Tinted-CIFAR10 | Skewed-CelebA, Waterbirds | TinyImageNet |
|---|---|---|---|
| ExLG | $lr = 0.1$ 
 $\lambda = 15.0$ | $lr = 0.03$ 
 $\lambda = 15.0$ | $lr = 0.03$ 
 $\lambda = 100.0$ |
| ImLG | $lr = 0.003$ | $lr = 0.0001$ | $lr = 0.003$ |

Table 8: Hyperparameters for continual learning analyses: All tasks are trained for 50 epochs with SGD optimizer for ExLG method and AdamW for ImLG.

| Method | Seq-CIFAR10 | Seq-TinyImageNet | DN4IL |
|---|---|---|---|
| ExLG | $lr = 0.05$ 
 $\lambda = 50.0$ | $lr = 0.05$ 
 $\lambda = 100.0$ | $lr = 0.03$ 
 $\lambda = 100.0$ |
| ImLG | $lr = 0.001$ | $lr = 0.0001$ | $lr = 0.0001$ |

architecture trained on the DN4IL dataset without any linguistic context. This model serves as a benchmark to assess the impact of language integration. ExLG (Explicit Language Guidance) is trained by integrating explicit linguistic descriptions during the training phase. An example of a description for "airplane" class is provided in Figure 1. This approach aims to align the visual encoder with semantic information directly tied to the visual content, promoting better generalization across varied domains. ImLG (Implicit Language Guidance) builds upon the Base model by incorporating a pretrained language model on top of the visual encoder. The ExLG model shows particularly notable enhancements in more challenging domains such as paintings and infographics, where traditional visual features are often insufficient for reliable recognition. The ImLG model also improves performance, but its effect is more subtle and is examined further through activation maps.

## B.2 Implicit Approach - Analysis

The implicit language guidance approach was motivated by the question of whether a lightweight language block, when placed on top of a vision encoder, can act as an attention filter for vision features, enriching them with higher-level semantic understanding. Instead of explicitly aligning vision and language representations, this method introduces a language model as an additional processing layer that implicitly refines visual features through its attention mechanisms. To analyze its effects, we visualize activation maps at different stages of the model—before and after adding the language block. Figure 7 demonstrates how the implicit method alters the model's focus: the baseline model primarily attends to local textures or background regions, while the implicit approach shifts attention toward more semantically meaningful parts of the object.

Further, we investigate whether this effect persists when the vision encoder itself is a transformer, specifically using ViT-Tiny. Since vision transformers already utilize self-attention to process spatial information, it remains unclear whether an additional language-driven attention block would still provide meaningful refinements. Figure 7 shows that even in this setting, the LM block continues to enhance feature selectivity, helping the model attend to conceptually relevant structures.

## C Similarity Preserving Loss function

Our alignment of vision and language representations follows a distinct approach. Unlike contrastive losses commonly used in VLMs that rely on large datasets for effective convergence, we adopt a knowledge distillation-inspired method using a similarity-preserving loss (Tung & Mori, 2019) to guide the image encoder with insights from the language model. Originally developed for a student-teacher framework, this loss

builds on the principle that semantically similar inputs elicit similar activation patterns in trained neural networks. In a knowledge distillation setting, the goal is for the trained teacher network to provide additional supervision to train a student network effectively.

In our framework, this loss (in Equation 2) ensures that inputs with similar semantic meanings in the language model induce correspondingly similar activations in the vision encoder, thereby fostering a shared representation space. By leveraging pre-trained language embeddings as a reference, the similarity-preserving loss enables the vision encoder to learn high-level, semantically rich features that transcend superficial correlations. Specifically, this loss supervises the vision encoder by comparing pairwise activation similarities within each batch and penalizing discrepancies in their similarity matrices. This approach bridges the textual and visual domains, enabling robust cross-modal learning with minimal additional training complexity.

## D    Experimentation setting

The summary of all the extensive analysis in the paper along with the corresponding datasets is shown in Table 1. As the pretrained Language model, we utilized Sentence Transformers (Reimers & Gurevych, 2019) and also CLIP text encoder (Radford et al., 2021). In both ExLG and ImLG experiments, we observed comparable performance across both models.

### D.1    Hyper-parameters

For all the baseline experiments, we adopt standard classification settings. For CIFAR-10, CIFAR-100, Tinted-CIFAR10, Skewed-CelebA and Waterbirds, we use a learning rate of 0.1 with SGD as the optimizer, training for 100 epochs. For TinyImageNet, we use a learning rate of 0.03 while keeping the same number of epochs and optimizer settings.

There is only one hyper-parameter for ExLG ($\lambda$) and no additional parameters for ImLG. The hyper-parameters for Explicit and Implicit methods are provided in The Tables 7 and 8.

**ImLG Architecture Details.** The frozen language block *lb* is the transformer encoder of LM - Sentence Transformer (all-MiniLM-L6-v2), with hidden dimension $d_{\text{llm}} = 384$ and all weights are frozen throughout training. The projection layers map between the visual and language feature spaces. The ResNet `layer4` output (shape $B \times 512 \times H \times W$) is flattened to $B \times d_{\text{emb}}$. The input projection $p_1 \colon \mathbb{R}^{d_{\text{emb}}} \to \mathbb{R}^{384}$ maps this to the LM hidden dimension. Each sample is treated as a single token and the projected vector is changed to $B \times 1 \times 384$, passed through all encoder, and reduced back to $B \times 384$. The output projection $p_2 \colon \mathbb{R}^{384} \to \mathbb{R}^{512}$ maps back to the visual feature dimension for the classifier. The block is inserted after `layer4`; only $p_1$, $p_2$, the backbone, and the classifier head are updated during training.

### D.2    IID and OOD Generalization

For the IID (Independent and Identically Distributed) setting, we evaluate on CIFAR-10, CIFAR-100, Tiny-ImageNet and ImageNet100 which are standard datasets used in classification tasks. To explore shortcut learning, we employ Tinted-CIFAR10 and Skewed-CelebA datasets, which introduce biases and distribution shifts designed to test the model's ability to avoid learning spurious correlations.

We conduct out-of-distribution (OOD) tests by training the model on TinyImageNet and testing it on three challenging OOD datasets: ImageNet-A, ImageNet-O, and ImageNet-R. ImageNet-A (Adversarial) (Hendrycks et al., 2021b) consists of naturally occurring adversarial examples that are misclassified by models trained on ImageNet, making it an ideal dataset for evaluating a model's adversarial robustness. ImageNet-O (Outliers) (Hendrycks et al., 2021b) contains outlier images that do not belong to any of the ImageNet classes, allowing us to test the model's ability to handle inputs outside of its training distribution. Lastly, ImageNet-R (Renditions) (Hendrycks et al., 2021a) includes artistic renditions of ImageNet classes, such as paintings, cartoons, and sculptures, which introduce significant style variations and help in evaluating the model's capacity for generalization across different visual domains.

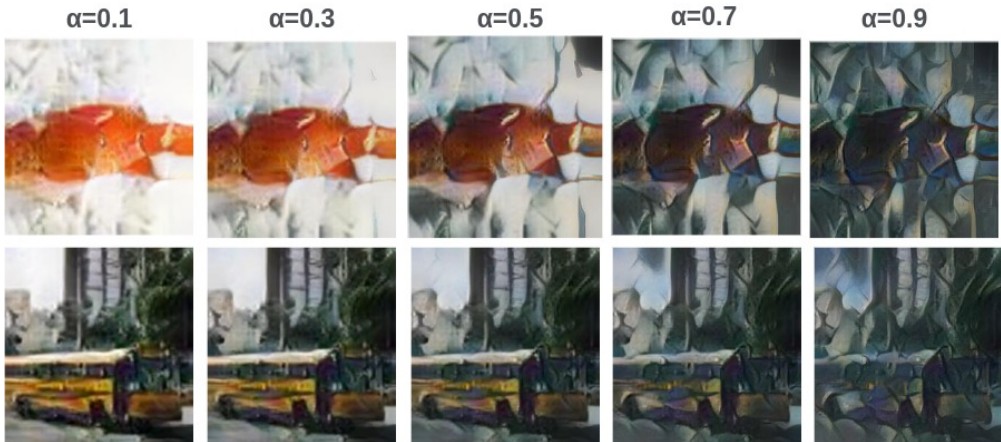

Figure 8: Few examples of Stylized-TinyImageNet dataset used in Texture vs Shape Analysis

## D.3 Shortcut Learning

**Tinted-CIFAR10**: Tinted-CIFAR10 is a modified version of the standard CIFAR-10 dataset, where each class is assigned a distinct color tint. This transformation introduces an artificial correlation between class labels and background color, creating a strong bias that can be exploited by the model. Since the actual object features remain unchanged, a model that relies primarily on the color tint for classification will struggle when presented with test images where the tint is removed or altered.

**Skewed-CelebA**: The Skewed-CelebA dataset is derived from the CelebA dataset, which contains a diverse set of face images annotated with multiple attributes. In this variation, we do a binary gender classification task and an artificial bias is introduced by heavily skewing the training data distribution: the majority of blonde individuals are female, while most non-blonde individuals are male. This creates a scenario where hair color becomes a spurious cue for gender classification. At test time, the dataset is designed to challenge the model by evaluating it on groups that were rare or unseen during training—blonde males and non-blonde females. A model that overfits to the spurious correlation between hair color and gender will perform poorly on these underrepresented groups, revealing its susceptibility to shortcut learning.

**Waterbirds**:The Waterbirds dataset leverages images from the CUB dataset (Wah et al., 2011), featuring various bird species, superimposed onto backgrounds from the Places dataset (Zhou et al., 2017), resulting in a combination that introduces a spurious correlation challenge in the data. The key variable in this dataset is the type of bird (landbird or waterbird), which serves as the target variable 'y'. Additionally, there is a spurious attribute 's' that indicates the type of background (land or water) each bird is placed against. This correlation is artificial; for example, while waterbirds are predominantly photographed against water backgrounds, this is not universally true in natural settings. The dataset explicitly defines four groups based on combinations of the bird type and the background type to investigate and mitigate the potential bias induced by this spurious correlation. This setup provides a rich framework to study and develop models that can differentiate between core features and misleading attributes that do not inherently define the class but are associated due to dataset biases.

## D.4 Continual Learning

In the continual learning setting, we explore Class-Incremental Learning (Class-IL) and Domain-Incremental Learning (Domain-IL), both of which are common benchmarks for evaluating continual learning models. In Class-IL, each task introduces new classes, and the model is required to learn these new classes while retaining knowledge of previously learned classes without forgetting. Task-IL is similar to Class-IL but has task identities to select the relevant classifier for each sample. In contrast, Domain-IL involves tasks where the class labels remain the same across tasks, but the input data distribution shifts with each new task. For Domain-IL, we focus on the DN4IL dataset.

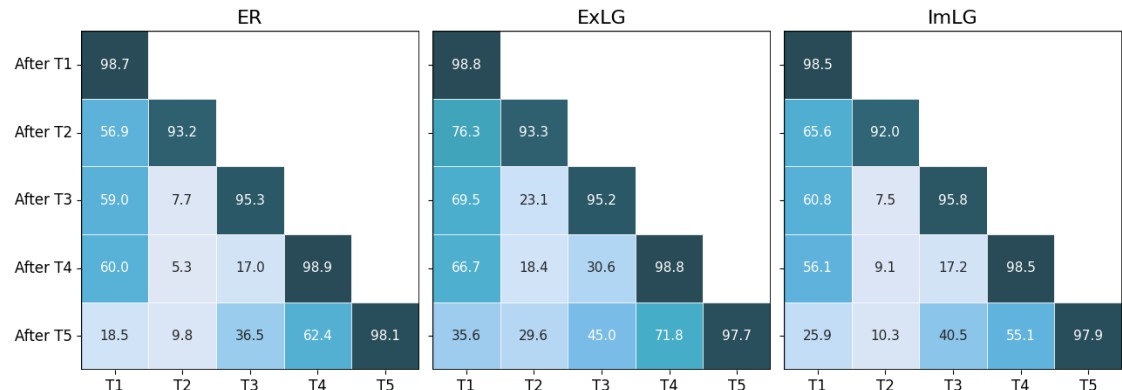

Figure 9: Task-wise performance of class-incremental learning setting on Seq-CIFAR10 dataset with 200 buffer size.

Table 9: Results with a bigger CNN backbone- ResNet50

|  | CIFAR10 | | | CelebA | | |
|---|---|---|---|---|---|---|
|  | Base | ExLG | ImLG | Base | ExLG | ImLG |
| ResNet50 | 94.38 | **97.21** | **95.86** | 63.88 | **74.45** | **76.58** |

The DN4IL (DomainNet for Domain-IL) dataset (Gowda et al., 2023) is a curated subset of the DomainNet dataset (Peng et al., 2019), originally used for domain adaptation tasks. It has common objects across six diverse domains: real, clipart, infograph, painting, quickdraw, and sketch. DN4IL offers a more succinct, balanced, and computationally efficient version of DomainNet, making it well-suited for benchmarking continual learning methods while preserving the challenging distribution shifts between domains.

Plasticity measures the model's capability to learn new tasks. It is calculated as the average accuracy of each task when it is first learned. For example, this is the accuracy of the network trained on task $T_2$, evaluated on the test set of $T_2$. Stability measures the model's ability to retain knowledge from previously learned tasks. It is computed as the average accuracy of all tasks from 1 to $T - 1$ after learning the final task $T$. Trade-off - To assess the balance between plasticity and stability, we use the following metric:

$$\text{Trade-off} = \frac{2 \times \text{Plasticity} \times \text{Stability}}{\text{Plasticity} + \text{Stability}}$$

Figure 9 shows the task-wise performance after each task on Seq-CIFAR10 dataset. The last row specifically shows the accuracy on all the tasks after the model finishes learning the final task. For example, the accuracy on Task 1 drops from 98.7 to 18.5 by the time the model finishes learning Task 5.

# E Additional Results and Ablations

## E.1 Different Image Encoders

In this section, we present additional results, starting with an evaluation of different vision encoders across various datasets. Table 9 provides results for both the ExLG and ImLG methods using the ResNet50 vision encoder. As the dataset complexity and vision model size scale up, we observe more significant improvements, demonstrating the scalability of our methods.

Table 10: Results with different language models using ExLG method.

| | | Cls | | Shortcut | OOD | | | CL |
|---|---|---|---|---|---|---|---|---|
| | | CIFAR10 | TinyImg | CelebA | ImgNet-O | ImgNet-R | ImgNet-A | DN4IL |
| | Base | 94.84 | 58.73 | 61.28 | 41.73 | 10.59 | 1.92 | 24.15 |
| ExLG | LM | 95.12 | 65.63 | 72.11 | 46.70 | **14.95** | **2.94** | **27.71** |
| | Larger LM | 95.01 | **65.89** | **72.93** | **47.51** | 14.65 | 2.92 | 26.84 |
| | MM-CLIP | **95.13** | 64.62 | 70.50 | 45.56 | 12.58 | 2.50 | 25.54 |

Table 11: Results with different language models using ImLG method.

| | | Cls | | Shortcut | OOD | | | CL |
|---|---|---|---|---|---|---|---|---|
| | | CIFAR10 | TinyImg | CelebA | ImgNet-O | ImgNet-R | ImgNet-A | DN4IL |
| | Base | 94.84 | 58.73 | 61.28 | 41.73 | 10.59 | 1.92 | 24.15 |
| ImLG | LM | 93.41 | 60.02 | 75.90 | **42.20** | 12.10 | 2.37 | 24.22 |
| | Larger LM | **93.78** | **61.16** | **77.55** | 42.12 | **12.71** | **2.45** | **24.51** |
| | MM-CLIP | 93.33 | 57.87 | 70.50 | 39.10 | 10.80 | 2.06 | 24.00 |

## E.2 Different Language Models

In this section, we evaluate the impact of different language models (LMs) on our framework. In the experiments presented in the main paper, we utilize the LM - Sentence Transformer (all-MiniLM-L6-v2) (Reimers & Gurevych, 2019), an efficient model with only 22.7M parameters, which adds minimal computational overhead. To further investigate, we conduct experiments using a Larger LM - "all-distilroberta-v1" with 82.1M parameters. Further, we also examine results with a language model that has been trained with multi-modality, such as the language encoder in CLIP (MM-CLIP).

Tables 10 and 11 show the results across various tasks using these two language models. While larger models yield improved performance, efficient models like all-MiniLM-L6-v2 are sufficient, provided that the descriptions are semantically rich.

Table 13 evaluates how the descriptions affect ExLG performance. We compare variants that progressively remove semantic content: replacing the pretrained LM with random weights (LM-Rand), substituting language embeddings with an identity similarity target ($\mathcal{S}_l = \mathbf{I}$), and using just class names only or deliberately misassigned descriptions. On all the datasets (IID, shortcut learning and continual learning), richer and correctly-paired descriptions yield better performance than class names alone, which in turn outperform wrong or random targets. This indicates that ExLG's gains are driven by the semantic content of the language supervision, not by the regularization effect of the alignment loss alone.

Table 12: Resource analyses.

| | | | CelebA | |
|---|---|---|---|---|
| | | Acc | #Trainable Params (M) | Training Time (seconds) |
| ResNet18 | Base | 61.28 | 11.17 | 575.63 |
| | Ex-LG | **72.11** | 11.17 | 590.91 |
| | Im-LG | **75.90** | 14.51 | 586.54 |
| ResNet50 | Base | 63.88 | 23.50 | 4100 |

Table 14 decomposes the ImLG gain into its constituent components. The projection-only variant retains the two linear layers $p_1$ and $p_2$ but removes the transformer block entirely, isolating the contribution of dimensional reshaping. LM-Rand adds a randomly-initialized frozen block of the same architecture, capturing any structural inductive bias from the transformer without pretrained knowledge. The weight-shuffled variant permutes the pretrained weights tensor-by-tensor. The results show that each component contributes incrementally. This confirms that it is the pretrained language knowledge encoded in the frozen block, rather than the projection layers or the transformer architecture, that drives ImLG's performance gains.

Table 13: ExLG ablation: effect of description quality on classification accuracy.

| ExLG Variant | CIFAR-10 | CelebA | DN4IL |
| --- | --- | --- | --- |
| Baseline | 94.84 | 61.28 | 24.15 |
| LM-Rand (random LM weights) | 92.17 | 67.24 | 22.08 |
| Shuffled descriptions | 93.05 | 67.85 | 24.78 |
| Simple (class names only) | 94.92 | 70.25 | 26.81 |
| **ExLG** | **95.12** | **72.11** | **27.71** |

Table 14: ImLG ablation: projection pipeline, transformer architecture, and pretrained language knowledge. It is reported on CelebA as ImLG shows the most imapct on shortcut learning datasets.

| ImLG Variant | CelebA |
| --- | --- |
| Baseline | 61.28 |
| Projection-only (no LM block) | 63.24 |
| LM-Rand (random frozen transformer block) | 67.41 |
| Weight-shuffle (permuted pretrained weights) | 70.00 |
| **ImLG** | **75.90** |

