# OpenReview forum: "Language Guidance for Supervised Vision Training: An Empirical Study of Generalization"
_TMLR — Under review for TMLR_

### Review · Reviewer_fJ5a · 2026-06-07

**Summary Of Contributions:**

This paper studies whether frozen pretrained language models can provide lightweight guidance for standard supervised vision training. The authors consider a setting where the training data consist only of image-label pairs. They propose two forms of language guidance: Explicit Language Guidance (ExLG), which uses class-level language descriptions to impose a similarity-preserving alignment loss on visual representations, and Implicit Language Guidance (ImLG), which projects visual features through a frozen language-model block before classification. The paper evaluates these approaches across several generalization regimes.

**Strengths**

1. The question is interesting and practically relevant: whether language-derived structure can improve supervised vision training without requiring a full vision-language training pipeline. I find the intuition behind ExLG plausible, since class-level language descriptions may encode semantic relations that are absent from one-hot labels.
2.  The empirical scope is also broad, and the paper reports improvements across a variety of datasets and settings. The comparison between explicit and implicit forms of language guidance is potentially useful.

**Weaknesses**

1.  For ExLG, the similarity-preserving objective may act as a generic regularizer or class-level structure prior, and the paper does not include sufficient controls such as random embeddings, shuffled class-description assignments, learned class embeddings, or non-semantic similarity targets.
2.  For ImLG, the mechanism is less conceptually clear: projecting visual features into a frozen language block does define an additional nonlinear transformation, but it is not obvious that such a block performs meaningful language-derived semantic filtering when its inputs are projected visual features rather than text-token representations.

**Audience:**

Yes

**Audience Explanation:**

Researchers in vision-language learning, robustness/generalization, and continual learning may be interested in this paper.

**Broader Impact Concerns:**

No additional broader impact concerns.

**Claims And Evidence:**

No

**Claims Explanation:**

The results support the narrower claim that ExLG and ImLG can improve over a supervised baseline in several settings. However, they do not convincingly support the stronger claim that these gains come specifically from language semantics.

For ExLG, the alignment loss may simply act as a generic regularizer or class-level structure prior. The paper lacks key controls such as random embeddings, shuffled text-class assignments, learned class embeddings, or non-semantic similarity targets.

For ImLG, the mechanism is not well justified. Projecting visual features through a frozen language block adds a nonlinear transformation, but it is unclear why this should perform semantic language guidance on non-text inputs. Controls such as a frozen random block, projection-only baseline, or architecture-matched non-language block are needed.

**Requested Changes:**

Add controls for ExLG to show that the gains come from language semantics rather than a generic similarity regularizer. At minimum, compare against random class embeddings, shuffled class-description assignments, learned class embeddings, class-name-only embeddings, and non-semantic similarity targets.

Add controls for ImLG to justify the claim of implicit language guidance. At minimum, compare against a frozen random transformer block, a projection-only baseline, a reinitialized or weight-shuffled language block, and an architecture-matched non-language block.

Clarify the ImLG architecture in detail: which language model/block is used, how visual features are projected into the block, whether the input is treated as one token or multiple tokens, what dimensions are used, and where the block is inserted.

Address optimizer and hyperparameter fairness. Since baseline/ExLG and ImLG use different optimizers, the paper should include optimizer-matched comparisons or justify why this does not affect the conclusions.

---

> ### Author Response · Authors · 2026-07-17
>
> We thank the reviewer for finding our work practically relevant and for the detailed feedback. We have made changes in paper (all changes in BLUE) and also address the questions below
>
> - ExLG experiments - also in Table 13 in Appendix. We ran on IID (Cifar10), shortcut learning (CelebA) and continual learning (DN4IL) datasets.
>
> | ExLG variant | CIFAR10 | CelebA | DN4IL |
> |---|---|---|---|
> | Baseline | 94.84 | 61.28 | 24.15 |
> | LM-Rand (random LM weights)  | 92.17 | 67.24 | 22.08 |
> | Shuffled descriptions | 93.05 | 67.85 | 24.78 |
> | Simple (class names only) | 94.92 | 70.25 | 26.81 |
> | **ExLG** | **95.12** | **72.11** | **27.71** |
>
> Passing text through a randomly-weighted LM (LM-Rand) produces unstructured embeddings; performance falls below baseline on CIFAR10 and DN4IL, with a modest gain on CelebA (only 2 classes). Shuffling descriptions across classes performs worse than using simple class-name sentences, confirming that correct pairing matters — rich text alone is insufficient. Simple (class names only) improves over shuffled descriptions but falls below full ExLG , showing that description richness contributes a further graded benefit. ExLG with full GPT descriptions achieves the best result on all three datasets.
>
> - ImLG experiments   - also in Table 14 in Appendix.
> We focus on CelebA as CIFAR10 is near ceiling for all methods and is not discriminative, especially in ImLG.
>
> | ImLG variant | CelebA |
> |---|---|
> | Baseline | 61.28 |
> | Projection-only (no LM block) | 63.24 |
> | LM-Rand (random frozen block)  | 67.41 |
> | Weight-shuffle (permuted pretrained weights) | 70.00 |
> | **ImLG** | **75.90** |
>
> Retaining only the two linear projections $p_1$ and $p_2$ with no transformer block (Projection-only) provides a small gain over baseline, attributable to the change in feature dimensionality alone. Inserting a randomly-initialised frozen block of the same architecture (LM-Rand) adds a further increment, indicating the transformer structure carries some inductive bias even without trained weights. Permuting the pretrained weights tensor-by-tensor (Weight-shuffle) preserves tensor shapes and norms but disrupts the trained organisation; it outperforms LM-Rand but falls well below pretrained ImLG. The largest gain comes from the pretrained ImLG, where the language knowledge encoded during pretraining is preserved. Each component contributes, and the pretrained weights account for the dominant share of the gain.
>
> - ImLG architecture details -
>
> We have added a detailed description to Section 3.4 and the appendix in the revision. The frozen block $lb$ is the transformer encoder of all-MiniLM-L6-v2 (hidden dim $d_\text{llm}=384$), with all weights frozen throughout training. The ResNet layer output (B×C×H×W) is flattened to B×embed\_dim, where embed\_dim is the flattened spatial size at the chosen insertion point. (Ex: 8192 for CIFAR10 (layer4: 512×4×4) and 73728 for CelebA (layer4: 512×12×12)). The input projection $p_1$: Linear(embed\_dim, 384) maps this to the LM hidden dimension. Each sample is treated as a single token — the projected vector is expanded to B×1×384 and passed through the frozen encoder; the output is collapsed back to B×384. The output projection $p_2$: Linear(384, 512) maps back to the visual feature dimension, feeding directly into the classifier head. The block is inserted after ResNet layer4 and before global average pooling; for layer2/layer3 insertion points, $p_2$ maps back to spatial dimensions so remaining ResNet layers continue before pooling.
>
> - Optimizers
>
> We ran optimizer-matched comparisons.
>
> | Method | Optimizer | CIFAR10 | CelebA |
> |---|---|---|---|
> | Baseline | SGD (paper) | 94.84 | 61.28 |
> | Baseline | AdamW | 92.35 | 58.72 |
> | ExLG | SGD (paper) | 95.12 | 72.11 |
> | ExLG | AdamW | 91.98 | 63.47 |
> | ImLG pretrained  | AdamW(paper) | 93.41 | 75.90 |
>
> AdamW *hurts* both Baseline and ExLG when applied to ResNet. ImLG uses AdamW not as an optimisation advantage but for better convergence with the frozen LM. The experiment shows the optimizer choice cannot explain ImLG's gains as AdamW alone, on the same ResNet backbone without the LM block, shows lower results.
>
> *Please let us know if you have more questions. Did we answer all the ExLG variants you had in mind, if the non-semantic targets meant something else, please let us know and we will incorporate that.   For ImLG, we treated "architecture-matched non-language block" as a randomly-initialised frozen transformer of the same architecture (LM-Rand above) — if you meant something else, please clarify and we will report new results. Thank you.*

---

### Review · Reviewer_Er9n · 2026-06-17

**Summary Of Contributions:**

This paper studies language guidance for supervised vision training. It proposes ExLG, which uses language-description similarity as an auxiliary training signal, and ImLG, which inserts a frozen language block into the visual feature pathway. ExLG outperforms the supervised baseline across several settings, including IID accuracy, low-data learning, OOD transfer, and continual learning. ImLG shows somewhat better results than ExLG in some shortcut-learning settings.

**Audience:**

Yes

**Audience Explanation:**

The finding that language-derived representation similarity can help supervised vision training is interesting to the TMLR audience. In particular, the ExLG results suggest that language similarity can provide useful structure beyond one-hot labels and may improve robustness, OOD generalization, and continual learning.

**Claims And Evidence:**

No

**Claims Explanation:**

The empirical results for ExLG are reasonably convincing, and I find this part of the paper clear. However, I do not think the paper sufficiently supports its broader claims about language guidance because ImLG is not well justified. The paper does not provide enough theoretical, mechanistic, or intuitive explanation for why passing projected visual features through a frozen language block should work as language guidance, and lead to better robustness.

I also find the choice of language model weak relative to the motivation. The paper relies mainly on a small SentenceTransformer model, while part of the motivation is based on broad claims about language/vision representation alignment and the Platonic Representation Hypothesis. Since scale is central to that hypothesis, it is unclear whether results based on such a small language model are enough to support the paper's broader framing.

**Requested Changes:**

The authors should either provide a much stronger justification for ImLG or remove it from the main paper. In its current form, ImLG is hard to understand as language guidance. The paper should explain, at least intuitively and experimentally, why a frozen language block should process projected visual features in a semantically meaningful way. If this cannot be justified, I think the paper would be clearer and stronger by focusing on ExLG.

The authors should strengthen the language model analysis. Since the paper invokes language/vision representation alignment and the Platonic Representation Hypothesis, it should evaluate stronger language models closer to those used in that line of work. Ideally, the paper should include an analysis of how performance changes with language model scale.

---

> ### Author Response · Authors · 2026-07-17
>
> We thank the reviewer for finding our paper interesting and for the questions.
>
> - We want to clarify about the paper's contribution again.
> The goal was not to propose a novel or incremental method, but to conduct a holistic empirical study of how language can be introduced into supervised visual training and how different modes of language guidance affect different forms of generalisation. ExLG and ImLG represent two fundamentally different design philosophies: ExLG introduces language explicitly, as a semantic alignment signal in the loss; ImLG introduces it implicitly, as a structural prior embedded in the model's feature processing pipeline. A paper that claims to study "how language guidance affects generalisation" must evaluate both modes — otherwise it characterises only half the design space.
> ImLG It is included because it reveals a different and complementary pattern of behaviour:
>
> | Setting | ExLG | ImLG | Winner |
> |---|---|---|---|
> | Clean accuracy (CIFAR10) | **95.12** | 93.41 | ExLG |
> | Shortcut resistance (CelebA) | 72.11 | **75.90** | ImLG |
> | Continual learning (DN4IL) | **27.71** | 24.22 | ExLG |
>
> ExLG is stronger at clean accuracy and continual learning. ImLG is stronger at shortcut resistance — precisely the setting where the visual signal is misleading and an externally-imposed structural prior is most beneficial. These complementary failure modes are *the finding*. Removing ImLG would replace a rich characterisation of the design space with a narrower, less informative picture.
>
> *New experiments* We ran the following variants on CelebA, where ImLG is discriminative (CIFAR10 is near ceiling and does not differentiate the methods):
>
> | ImLG variant | CelebA |
> |---|---|
> | Baseline | 61.28 |
> | Projection-only (no LM block) [new] | 63.24 |
> | LM-Rand (random frozen block) [new] | 67.41 |
> | Weight-shuffle (permuted pretrained weights) [new] | 70.00 |
> | **ImLG** | **75.90** |
>
> Each component contributes a measurable gain, but the largest step comes from pretrained ImLG.
>
> *The mechanistic explanation:* A pretrained transformer encoder block encodes structural relationships in its attention and FFN weights — relationships shaped by processing vast amounts of semantically related text. When visual features are projected into the LM's hidden space and passed through this block, they are reorganised according to the LM's learned structural prior.
>
> - LM scale
>
> We take this point and will revise accordingly. Tables 10/11 (already in the Appendix)  provide comparison with 3 LM models: all-MiniLM-L6-v2 (22.7M), all-distilroberta-v1 (82.1M), and CLIP text encoder (~150M)
>
> **Table 10: Results with different language models using ExLG**
>
> | | CIFAR10 | TinyImg | CelebA | ImgNet-O | ImgNet-R | ImgNet-A | DN4IL |
> |---|---|---|---|---|---|---|---|
> | Base | 94.84 | 58.73 | 61.28 | 41.73 | 10.59 | 1.92 | 24.15 |
> | ExLG (LM) | 95.12 | 65.63 | 72.11 | 46.70 | **14.95** | **2.94** | **27.71** |
> | ExLG (Larger LM) | 95.01 | **65.89** | **72.93** | **47.51** | 14.65 | 2.92 | 26.84 |
> | ExLG (MM-CLIP) | **95.13** | 64.62 | 70.50 | 45.56 | 12.58 | 2.50 | 25.54 |
>
> **Table 11: Results with different language models using ImLG**
>
> | | CIFAR10 | TinyImg | CelebA | ImgNet-O | ImgNet-R | ImgNet-A | DN4IL |
> |---|---|---|---|---|---|---|---|
> | Base | 94.84 | 58.73 | 61.28 | 41.73 | 10.59 | 1.92 | 24.15 |
> | ImLG (LM) | 93.41 | 60.02 | 75.90 | **42.20** | 12.10 | 2.37 | 24.22 |
> | ImLG (Larger LM) | **93.78** | **61.16** | **77.55** | 42.12 | **12.71** | **2.45** | **24.51** |
> | ImLG (MM-CLIP) | 93.33 | 57.87 | 70.50 | 39.10 | 10.80 | 2.06 | 24.00 |
>
> Larger LMs yield further gains, and MM-CLIP (multimodal models ) is not always best — consistent with the finding that semantic richness matters more than model scale.
>
> Please let us know if anything needs more clarification or if you would like results with a different LM. Thank you.

---

### Review · Reviewer_SdWo · 2026-07-06

**Summary Of Contributions:**

the paper proposes a practical way to inject semantic structure into vision training and tests it across many generalization problems.

**Audience:**

Yes

**Audience Explanation:**

The paper discusses induction of semantic information into the training process, which is very relevant for large scale models.

**Claims And Evidence:**

Yes

**Claims Explanation:**

the paper provides a lot of empirical results supporting their claims, though some of the claims need more experiments/ablations, pl see below.

**Requested Changes:**

The paper presents interesting empirical results and is largely well written. i have minor concerns that i hope the authors can address :-

(1) a mechanistic or even an intuitive explanation of "why". would itbe a good idea to measure changes in representation geometry e.g. interclass margins, within class variance, or "entanglement" metrics (from unlearning literature) ?

(2) some ablations on sensitivities of text explanations should add to the paper's conclusions e.g. "winged animal" rather than "bird"

(3) the implicit architecture feels adhoc (also connected to "why" above) -- did the authors try other alternatives ?e.g. using a random frozen transformer block vs pretrained blocks, may be different insertion layers? do the results hold over multiple seeds robustly ?

(4) the paper is broad but feels a bit shallow. it is an interesting starting point and i wonder if the authors can comment on potential impact on future directions such as (not limited to) continual or multi modal learning. actually i think multimodal settings are perfect candidates for implicity guidances.

---

> ### Author Response · Authors · 2026-07-19
>
> We thank Reviewer for the thoughtful and constructive feedback. We address each point below.
>
> - Motivation -
>     -  ExLG - The similarity-preserving loss (Eq. 4) penalises any deviation between the visual inter-class similarity matrix $\mathcal{S}_v$ and the LM-derived semantic similarity matrix $\mathcal{S}_l$. This acts as a semantic regularizer and anchors the visual feature space to a structure that reflects language class relationships. The motivation is well-supported in the literature and Knowledge distillation from richer similarity targets (rather than hard labels) is known to improve generalization [1]. More broadly, aligning visual representations with language-derived class structure has been shown to improve transfer across datasets and settings [2].
>     - ImLG - In parameter-efficient fine-tuning, frozen pretrained transformer layers are inserted as adapters that reorganise representations without changing their semantics. Recent multimodal architectures ( BLIP, LLaVA..) show that a small trainable interface between a frozen vision encoder and a frozen LM is sufficient to align modalities and our approach is the reverse direction, using the LM structure to reorganise visual features.  We also show the activation maps in Figure 7 in Appendix.
>
> Regarding representation geometry, Our paper already includes a **CKA-based representation alignment analysis** (Appendix B.1) on the DN4IL dataset, which spans six visually diverse domains: real photos, clipart, infograph, paintings, sketches, and quickdraw. A vision-only baseline shows low cross-domain feature alignment, especially for visually challenging domains such as paintings, sketches, and quickdraw. Both ExLG and ImLG increase this cross-domain similarity, suggesting that language guidance anchors visual representations to semantic content that persists across domain appearance shifts, reducing reliance on visual surface statistics
>
> We have a question to clarify: when you suggest measuring inter-class margins and within-class variance, do you mean extracting features from trained model checkpoints (baseline, ExLG, ImLG) and computing class-level statistics (e.g., mean pairwise distance between per-class centroids, mean intra-class spread)?  Please clarify and we will run the analysis and report it (as we already have checkpoints).
>
> - Descriptions - We ran description ablations across three datasets - IID (Cifar10), shortcut learning (CelebA) and Continual learning (DN4IL)
>
> | Description type | CIFAR10 | CelebA | DN4IL |
> |---|---|---|---|
> | Baseline (no language guidance) | 94.84 | 61.28 | 24.15 |
> | LM-Rand (random LM weights)  | 92.17 | 67.24 | 22.08 |
> | Shuffled descriptions  | 93.05 | 67.85 | 24.78 |
> | Simple (class names only) | 94.92 | 70.25 | 26.81 |
> | **ExLG ** | **95.12** | **72.11** | **27.71** |
>
> Thus description quality matters and the benefit is graded richer, correctly-paired semantic content leads to better generalization.
>
> - ImLG - We focus on CelebA, where ImLG is most effective. We ran three alternative variants
>
> | ImLG variant | CelebA |
> |---|---|
> | Baseline | 61.28 ± 1.21|
> | LM-Rand (random frozen block) | 57.41 ± 8.1 |
> | **ImLG** | **75.90 ± 1.8** |
>
> ImLG pretrained is consistent across seeds. LM-Rand is unstable and we ran many seeds and few seeds collapsed to near-chance (38.65) and few achieved modest gains (67.41).
>
> *Insertion layer ablation:* We ran all three insertion depths on CelebA.
>
> | Insertion point |  Mean ± std |
> |---|---|
> | layer2  | 69.09 ± 5.2 |
> | layer3 | 74.45 ± 2.3 |
> | layer4  | 75.90 ± 1.8 |
>
> Later layers are better aligned with the LM's input distribution than low-level (layer2, higher variance) layers.
>
> [1] Frederick Tung and Greg Mori. Similarity-preserving knowledge distillation.
> [2] Mert Bulent Sariyildiz, Julien Perez, and Diane Larlus. Learning visual representations with caption annotations.

---

> ### Author Response · Authors · 2026-07-19
> **2/2**
>
> - Future work
>
>     **Continual learning.** Our paper already demonstrates efficacy of ExLG in the continual learning setting, and is complementary to consolidation-based methods like EWC , which penalise parameter changes. ExLG instead penalises *representational drift* at the feature level.
>
>     **Multimodal learning** - The dominant paradigm for incorporating language into vision is contrastive pretraining on image-text pairs (CLIP, ALIGN). This is powerful but expensive: CLIP was trained on 400M image-text pairs with significant compute, and collecting paired data for specialised domains is often infeasible. Our approach requires *no paired visual-language data at all* and only class descriptions, which can be generated with an LLM. ExLG injects semantic structure through a loss term at negligible overhead; ImLG does so through a frozen block at inference cost of a single transformer forward pass. For practitioners who need language-informed visual representations but cannot afford contrastive pretraining or who work in domains where paired data is scarce our approach is a practical, low-cost proxy.
> This could also be used as a template for training any other modality as well and can work as a lightweight path to multimodal alignment.
>
> Please let us know if you have more questions. Thank you

---

### Author Response · Authors · 2026-07-19
**General Comment**

We thank all reviewers for all the careful and constructive engagement with our work.  We want to provide a general summary of our work and new experiments in rebuttal

- Goal - The paper asks a focused question: can off-the-shelf frozen language models provide useful supervision for supervised vision training, without any paired image-text data or contrastive pretraining?  We study this through two lightweight mechanisms and evaluate them comprehensively across in-distribution classification, shortcut learning, out-of-distribution generalization, and continual learning, characterising when and how each form of guidance helps.

- Motivation - Supervised vision training relies on one-hot labels, which identify class membership but provide no information about semantic attributes, relations between classes, or higher-level conceptual structure. As a result, vision encoders must infer all meaningful abstractions from pixel statistics alone, which can encourage reliance on spurious surface cues rather than semantically grounded representations. Language offers a natural source of this missing structure. We study two fundamentally different ways of incorporating this. ExLG introduces language explicitly using class descriptions. The LM is used only during training and adds no inference overhead. ImLG introduces language implicitly and a frozen pretrained LM block is inserted directly into the vision pipeline, allowing language-derived structure to influence visual features through the LM's internal transformations rather than through an explicit alignment objective. This distinction is motivated by cognitive theories separating explicit, verbalizable reasoning from implicit, structure-shaping processing (Kahneman, 2011): ExLG is a System 2 intervention deliberate, description-driven, class-specific while ImLG is a System 1 intervention structural, always-on, requiring no class descriptions at all.

- New results provided in this rebuttal : In response to reviewer feedback we ran and report the following new experiments:
    -  a description quality ablation across CIFAR10, CelebA, and DN4IL ;
    - ImLG component ablations isolating the contribution of the projection layers, transformer architecture, and pretrained weights;
    - multi-seed stability analysis for ImLG pretrained vs LM-Rand;
    - insertion layer ablation across layer2/3/4;
    - optimizer  experiments.
    - We also already had the results on all bigger LM models in Appendix.

- Impact -  Beyond the specific results, this work makes a methodological contribution to a broader question of whether pretrained models from one modality can serve as auxiliary sources of structure for learning in another, without full multimodal pretraining. Our work opens a practical direction for the community that In any supervised setting where one-hot labels are structurally weak or specialised domains with scarce paired data, a language model (or any modality model) can serve as a plug-in source of semantic inductive bias. ExLG requires only class descriptions (only during training) and ImLG requires nothing beyond inserting a frozen block. Neither requires paired image-text data, prompt engineering, contrastive pretraining.  This paper provides a principled empirical foundation for that broader research direction.